# Tandemly repeated NBPF HOR copies (Olduvai triplets): Possible impact on human brain evolution

Matko Glunčić[1] , Ines Vlahović[2], Marija Rosandić[3,4], Vladimir Paar[1,4]

Previously it was found that the neuroblastoma breakpoint family (*NBPF*) gene repeat units of ~1.6 kb have an important role in human brain evolution and function. The higher order organization of these repeat units has been discovered by both methods, the higher order repeat (HOR)-searching method and the HLS searching method. Using the HOR searching method with global repeat map algorithm, here we identified the tandemly organized NBPF HORs in the human and nonhuman primate NCBI reference genomes. We identified 50 tandemly organized canonical 3mer NBPF HOR copies (Olduvai triplets), but none in nonhuman primates chimpanzee, gorilla, orangutan, and Rhesus macaque. This discontinuous jump in tandemly organized HOR copy number is in sharp contrast to the known gradual increase in the number of Olduvai domains (NBPF monomers) from nonhuman primates to human, especially from ~138 in chimpanzee to ~300 in human genome. Using the same global repeat map algorithm method we have also determined the 3mer tandems of canonical 3mer HOR copies in 20 randomly chosen human genomes (10 male and 10 female). In all cases, we found the same 3mer HOR copy numbers as in the case of the reference human genome, with no mutation. On the other hand, some point mutations with respect to reference genome are found for some NBPF monomers which are not tandemly organized in canonical HORs.

## Introduction

### The ~1.6-kb repeat units in the neuroblastoma breakpoint family (NBPF) genes

*NBPF* genes in human chromosome 1 contain a pronounced repetitive structure of ~1.6-kb repeat units (Fortna et al, 2004; Vandepoele et al, 2005; Popesco et al, 2006), with sizable divergence (~20%) among neighboring repeat units. These repeat units encode the protein domain previously named DUF1220 domain (Vandepoele et al, 2005), which was changed to Olduvai domain (Sikela & van Roy,

2017). These repeat units within *NBPF* genes were previously also called NBPF repeats or NBPF monomers (Paar et al, 2011) (Fig 1), in accordance with repeat terminology used by Willard (1985) in studies of *α* satellite repeats. Each of Olduvai domains (NBPF repeat units) is characterized by a structure with two evenly spaced exons (Popesco et al, 2006; Paar et al, 2011). Comprehensive studies have shown that the Olduvai domain copy number was correlated with brain size, cortical neuron number, IQ scores, cognitive aptitude, evolution, and with brain pathologies (autism, schizophrenia, microcephaly, macrocephaly, and neuroblastoma) (Vandepoele et al, 2005, 2008; Popesco et al, 2006; Andries et al, 2012; Dumas et al, 2012; Davis et al, 2014; Keeney et al, 2014; Quick et al, 2016; Astling et al, 2017; Mitchell & Silver, 2018; Fiddes et al, 2019; Heft et al, 2020). O'Bleness et al (2012, 2014) showed that expanded Olduvai triplets are responsible for the extreme increase in Olduvai copy number in humans. The linking of Olduvai copy number and human brain evolution was first suggested in Popesco et al (2006).

### HLS-searching method

Comparing sequences of repeated Olduvai domains, it was found that they are predominantly of three types, which were referred to as HLS-1, HLS-2, and HLS-3, and most often they appear in triplets (O'Bleness et al, 2012; O'Bleness et al, 2014) (Fig 1). Because of recent change of terminology, which was fully justified (Sikela & van Roy, 2017), these HLS triplets were referred to as Olduvai triplets. HLS searching method identifies Olduvai domains in a given genomic sequence in the first step and compares their divergence to determine Olduvai triplets in the second step.

### Higher order repeat (HOR)-searching method

Centromeric regions of primate chromosomes are largely built from *α* satellite repeat units of of length ~171 bp, which have been extensively studied (Manuelidis, 1978; Willard, 1985; Jorgensen et al, 1987; Tyler-Smith & Brown, 1987; Waye &Willard, 1987; Choo et al, 1991; Jurka et al, 1996, 2005; Warburton & Willard, 1996; Alexandrov et al, 2001; Rosandic et al, 2003; Rudd & Willard, 2004; Paar et al, 2005; Warburton et al, 2008; Aldrup-Macdonald & Sullivan, 2014; Miga,

[1]Faculty of Science, University of Zagreb, Zagreb, Croatia   [2]Algebra University College, Zagreb, Croatia   [3]University Hospital Centre Zagreb (ret), Zagreb, Croatia   [4]Croatian Academy of Sciences and Arts, Zagreb, Croatia

Correspondence: matko@phy.hr

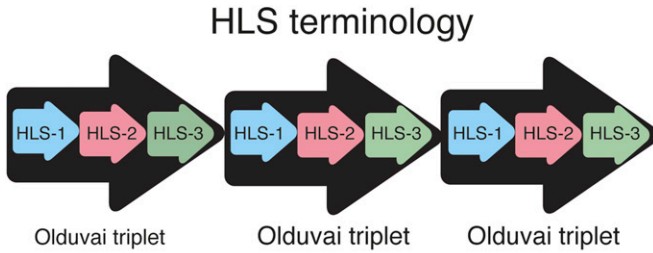

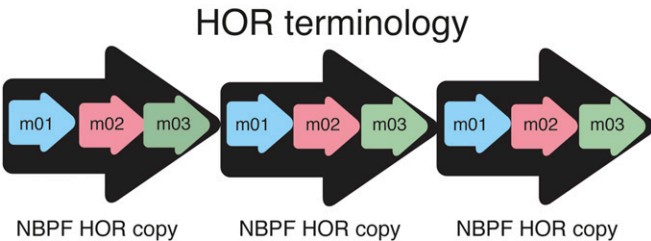

**Figure 1.  HLS versus higher order repeat terminology.**

2017; Sullivan et al, 2017; Lower et al, 2018; Gluncic et al, 2019; Uralsky et al, 2019). Neighboring α satellite monomers (primary repeat units) diverge sizably from each other (~20–40%), but some stretches of *n* monomers are often organized into secondary repeat *n*mer HOR copies, with small mutual divergence between HOR copies (less than 5%, in some cases even below 1%). Thus, divergence between HOR copies is much smaller than between monomers within each HOR copy (Warburton & Willard, 1996). Pronounced α satellite *n*mer HORs in human genome are chromosome specific, for example, 2mer and 11mer in chromosome 1; 15mer in chromosome 8; 5mer in chromosome 11, 12mer in chromosome X, and 34mer in chromosome Y (Alexandrov et al, 2001).

An efficient and robust computational tool to identify large HOR copies, as for example, α satellite HORs, is the global repeat map (GRM) algorithm (Paar et al, 2011; Gluncic & Paar, 2013; Gluncic et al, 2019; Vlahović et al, 2020). HOR method to determine α satellite repeat structure is based on identifying α satellite HOR copies in a given sequence in the first step, and in the second step deducing α satellite monomers from the HOR structure. Thus, HOR method enables identification of HOR copies without any prior knowledge of the primary repeats on which the HORs are based. Because the divergence among HOR copies is much smaller than the divergence among its underlying primary repeats, the scope of computations is simpler in the case of using HOR method than in the method of using monomer identification.

GRM algorithm identifies all pronounced HORs in a given genomic sequence. For example, applying in 2011 the GRM algorithm to the Builds 36.3 assembly of human chromosome 1 we obtained three HORs (Paar et al, 2011):

(1) The already known α satellite 11mer HOR (based on ~171-bp primary α satellite repeat unit);
(2) The novel 3mer HOR, based on previously known ~1.6-kb primary repeat units from NBPF genes in human chromosome 1;
(3) The novel quartic HOR, based on known 39-bp hornerin primary repeat unit in human chromosome 1.

Whereas the 11mer α satellite HORs are located in the centromere, the last two HORs are located within NBPF genes and Hornerin genes, respectively.

We also note that by applying GRM algorithm to genomic sequences of insects, the HOR pattern (based on T-cast-360 monomers of 331, 361, 362, and 369-bp primary repeat units), has been also discovered in beetle *Tribolium castaneum* (Vlahovic et al, 2017), which is evolutionary very distant from primates.

Using the robust GRM algorithm, we have first discovered in Build 36.3 assembly of human chromosome 1 the ~4,770-bp peak of 3mer HOR (Paar et al, 2011). The GRM diagram for this ~4,770-bp consensus sequence revealed two additional internal GRM peaks, at ~1.6 and ~3.2 kb. The GRM diagram for the 3.2-kb peak also shows that its consensus sequence consists of two ~1.6-kb monomer repeats. Altogether, the ~4,770-bp consensus sequence consists of a tandem of three ~1.6-kb monomer repeats of consensus lengths 1,623, 1,593, and 1,554 bp, respectively. The corresponding consensus monomers were denoted m01, m02, and m03, respectively. In this way, we discovered that the ~4,770-bp repeat copies are the 3mer HORs. Because these 3mer HORs are located within the *NBPF* genes in human chromosome 1, we named them NBPF 3mer HORs, and their constituent ~1.6-kb monomers were referred to as NBPF monomers in accordance with Willard's terminology for HORs (Warburton & Willard, 1996). Each consensus monomer consists of two exons and three introns. Each consensus monomer contains in succession intron-exon-intron-exon-intron. In the 1,554-bp consensus monomer their lengths amount to ~8%, ~4%, ~39%, ~12%, and ~37% of monomer length. In the 1,593-bp consensus monomer the length of second intron increases by ~8%, and in the 1,623-bp consensus monomer additionally the length of the first intron increases by ~6%.

In the Build 36.3 genomic assembly of human chromosome 1, we identified 47 HOR copies forming tandems, but only 34 of these 47 were canonical as shown in Fig 8 from Reference Paar et al (2011). The three constituent monomers of ~1.6 kb correspond to HLS DUF1220 domains of Fortna et al (2004), Vandepoele et al (2005), and Popesco et al (2006). As pointed out in Andries et al (2012), before the discovery of NBPF 3mer HOR by Paar et al (2011), it was not realized by HLS-searching method that the HLS domains form the triplet organization, that is, that the diverging DUF1220 HLS domains are of three types, forming a remarkable triplet organization—the HORs. On the other hand, the Build 2.1 (2010) genomic assembly of chimpanzee chromosome 1 did not show tandemly organized NBPF HOR copies, although 14 individual NBPF HOR copies (not organized into tandem) were identified and the total number of dispersed NBPF monomers was 48 (Paar et al, 2011). Also, tandemly organized NBPF HOR copies were not found in genomic assembly WUSTL Pongo_albelii-2.0.2 (2010) of orangutan and in Build 1.1 (2010) of Rhesus macaque (Paar et al, 2011). On this basis we hypothesized that the NBPF HOR tandem repeats (47 NBPF HOR copies in human versus 0 in nonhuman primate genome) reveal the NBPF HOR tandem repeats are human specific, possibly contributing to human brain evolution and human-chimpanzee divergence (Paar et al, 2011).

A possible challenge to such hypothesis has arisen from another complex HOR pattern, discovered by GRM algorithm within the hornerin (HRNR) gene in Build 36.3 genome assembly of human chromosome 1 (Paar et al, 2011): nine 39-bp primary repeat HRNR

units organized into ~0.35-kb secondary repeat units → two ~0.35-kb secondary HOR repeat units organized into ~0.7-kb tertiary HOR repeat units → and finally two 0.7-kb tertiary HOR repeat units organized into ~1.4-kb quartic HOR repeat units. In this, the 1,410-bp repeat unit represents the quartic HOR repeat unit, which is tandemly repeated, as the most complex multi-step HOR pattern discovered so far. In Reference Paar et al (2011), the quartic HOR was detected in the human genome only, and no quartic HOR counterpart was found in then-available sequenced chimpanzee genome.

The more recent investigation by Romero et al (2018) of HRNR from more complete NCBI human genomic assembly NC_000001.11 (2018) fully confirmed the HRNR quartic HOR formation described by Paar et al (2011). However, Romero et al (2018) found that the tandem quartic HOR formation of HRNR was conserved also in more recent reference genomes of chimpanzee and other nonhuman primates, except the crab-eating macaque. This showed that the tandemly organized HRNR quartic HOR copies are not exclusively human specific, as appeared for reference genome in 2011, but are also present in more recent NCBI reference genome sequences of most primates (Romero et al, 2018).

Prompted by that development, here we investigate whether the tandemly organized canonical NBPF 3mer HORs are present in current genome assemblies in both human and nonhuman primate genomes, or are exclusively human-specific, as was indicated by previous chimpanzee reference genome.

## Results

### Computed GRM diagrams for human genome assembly NC_000001.11 (2022) of human chromosome 1 and resulting NBPF HOR/monomer pattern

GRM diagram for NC_000001.11 (2022) human genome assembly of chromosome 1 shows a peak at ~4.8 kb, corresponding to the ~4.8 kb tandemly organized canonical NBPF 3mer HOR copies, based on ~1.6 kb-NBPF monomer (Fig 2A). Its constituents ~1.6-kb NBPF monomers contribute also to the GRM peaks at ~1.6 and ~3.2 kb.

Using GRM algorithm, we obtain that the arrays of tandemly organized canonical NBPF HOR copies are located in human genes *NBPF20* (19 canonical HOR copies), *NBPF19* (13 canonical HOR copies), *NBPF10* (11 canonical HOR copies), and *NBPF14* (7 canonical HOR copies), in total 50 tandemly organized canonical NBPF HOR copies. In addition, we identified 22 noncanonical NBPF HOR copies (Table S2). Canonical 3mer HOR copy is defined as follows: (i) each HOR copy is constituted of three monomers m1, m2, and m3, where each monomer diverges from its consensus by less than 5% and (iii) each HOR copy is in tandem with two or more HOR copies. Human consensus NBPF monomers, determined from canonical NBPF HORs, are displayed in Table S1.

The human NBPF HOR pattern with 50 canonical NBPF HOR copies encompasses 34 canonical HOR copies identified previously in NCBI (2010) genome assembly (Paar et al, 2011) and is similar to the HLS DUF1220 triplet repeat pattern identified by monomer searching method (O'Bleness et al, 2014; Heft et al, 2020).

Distribution of NBPF monomers among *NBPF* genes is shown in Fig 2, as obtained by HOR-searching method. Additional isolated NBPF monomer or HOR copies could be found by increasing divergence interval. For example, in the gene *NBPF12* a doublet of HOR copies, with one monomer missing in the second HOR copy, could change into a doublet of two canonical HOR copies by extending divergence limit for recognition of the third monomer. Analogously, an isolated noncanonical HOR copy could become canonical.

Divergence between canonical NBPF HOR copies is an order of magnitude smaller than divergence between monomers within each NBPF HOR copy. For example, in the NBPF20 array of 19 canonical NBPF HOR copies, the average divergence between NBPF HOR copies is ~1.6%, whereas the average divergence between monomers within each NBPF HOR copy is ~18%. This pattern is consistent with divergence pattern for the well-known α satellite HORs constituted from the 171-bp α satellite monomers (Warburton & Willard, 1996).

### Computed GRM diagrams and NBPF monomer distributions for genome assemblies of nonhuman chromosomes 1 of chimpanzee (NC_036879.1, 2022), gorilla (NC_044602.1, 2022), orangutan (NC_036903.1, 2022), and rhesus macaque (NC_041754.1, 2022), and resulting NBPF HOR/monomer pattern

GRM diagrams for nonhuman genome assemblies of the whole chromosome 1 are displayed in Fig 2B–E (left panel), which shows that the peak at ~4.8 kb, corresponding to the tandemly repeated NBPF 3mer HORs, is absent. This is even more evident in GRM diagrams for genome segments which contain NBPF genes (Fig 2B–E, right panel).

As an alternative method to search for distribution of NBPF monomers in nonhuman primate genomes, we applied Edlib (Sosic & Sikic, 2017), an open-source C/C ++ library for exact pairwise sequence alignment using edit (Levenshtein) distance, with human NBPF consensus monomers m1, m2, m3, obtained by GRM algorithm. Distribution and divergence of NBPF monomers, 3mer HORs and 2mer variants in primate genomes (human, chimpanzee, gorilla, orangutan, and Rhesus macaque) are displayed in Table S2. Distribution of NBPF monomers in nonhuman primate genomes shows the absence of tandemly repeated canonical NBPF HORs (Fig 3), in full accordance with the absence of the 4.8 kb GRM peak in GRM diagrams (Fig 2B–E).

Both methods (1) computing GRM diagram for each nonhuman primate genome and looking for existence of the ~4.8-kb GRM peak, and (2) using the human NBPF consensus monomers, determined by GRM algorithm, to analyze genomes of nonhuman primates, prove the absence of tandemly organized canonical NBPF HORs in nonhuman primate genomes.

### Comparison of copy number and structural organization of tandemly organized NBPF 3mer HOR copies of 20 randomly chosen individual human genomes from The International Genome Sample Resource and human reference genome GRCh38

Variations in Olduvai triplet copy number among various human genomes have been reported (Heft et al, 2020). To observe which is the diversity of this genomic portion in terms of copy number and

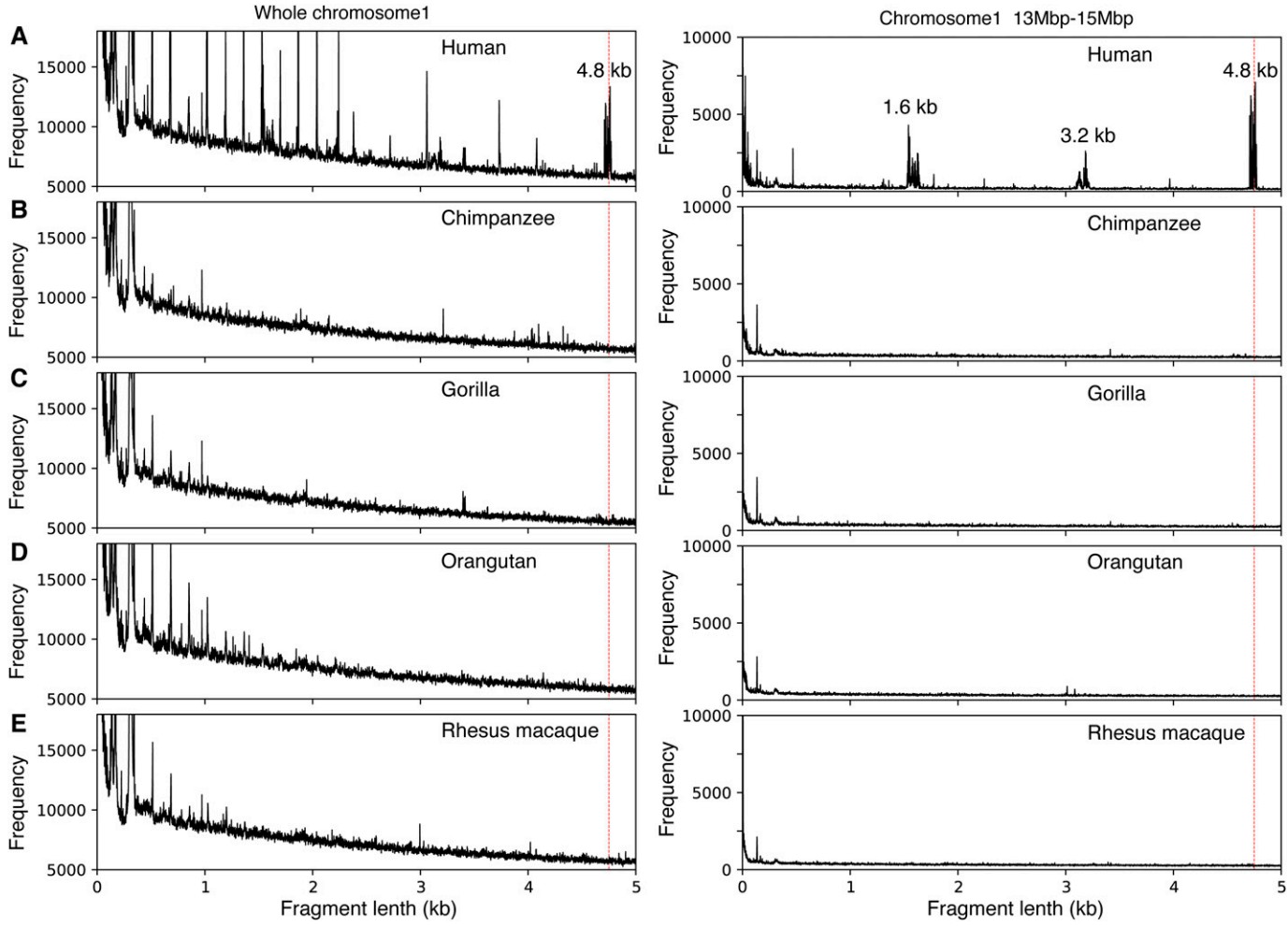

**Figure 2. Global repeat map diagrams for National Center for Biotechnology Information (2022) primate genome assemblies.**
**(A)** Human, NC_000001.11. **(B)** Chimpanzee, NC_036879.1. **(C)** Gorilla, NC_044602.1. **(D)** Orangutan, NC_036903.1. **(E)** Rhesus macaque, NC_041754.1. Peaks at ~4.8 kb correspond to the NBPF 3mer higher order repeat. These peaks are present in human and absent in nonhuman primate genomes. These diagrams demonstrate that the NBPF 3mer higher order repeat is exclusively characteristic of human genome, completely absent in National Center for Biotechnology Information (2022) assemblies of nonhuman primate genomes.

mutations in different genomes, we have applied our HOR-finding method to sequences of 20 randomly chosen individuals from The International Genome Sample Resource (The 1000 Genomes Project: http://www.internationalgenome.org/data-portal) (Table S3). In all 20 genomes, we obtained exactly the same copy numbers and the same consensus HOR sequences (with no mutations) equal to results for reference genome GRCh38, as illustrated by Table 1. (For full results on all 20 genomes, see Table S4, where monomers marked by asterisk do not belong to canonical 3mer HOR copies and some have a point mutation with respect to results for reference genome GRCh38.) In the table, the same value of divergence for the corresponding monomers in canonical HOR copies for all 20 individual genomes and reference genome means that monomers in canonical HOR copies in all 20 individual genomes have no mutation with respect to the reference genome. On the other hand, a sizable number of monomers from HOR copies outside of tandemly organized 3mer HOR copies have some point mutations with respect to the reference genome, although their HOR copy number

and structural organization (monomer types) are the same as for reference genome. For example, the every monomer No. 30 in 20 individual genomes from Table S3 has the same monomer length 1,529 bp and monomer type m01, but only eight of them have the same divergence value (14.54%) as the corresponding monomer No. 30 for reference genome (i.e., no point mutation with respect to the reference genome), whereas the remaining 12 individual genomes have slightly different divergences (14.60%, 14.67%) (i.e., point mutation with respect to the reference genome). Because of large length of NBPF monomer (~1.6 kb) such rare point mutations cannot influence the NBPF HOR structure and copy numbers, but are restricted only to slight difference of monomer sequences. In this sense, the tandemly repeated 3mer consensus HOR pattern may be considered as a kind of characteristic signature of human genome. A recent study indicates that human NBPF proteins that contain many tandemly repeated Olduvai triplets are posttranslationally cleaved by the furin protease once at each triplet (Pacheco et al, 2022 Preprint). This

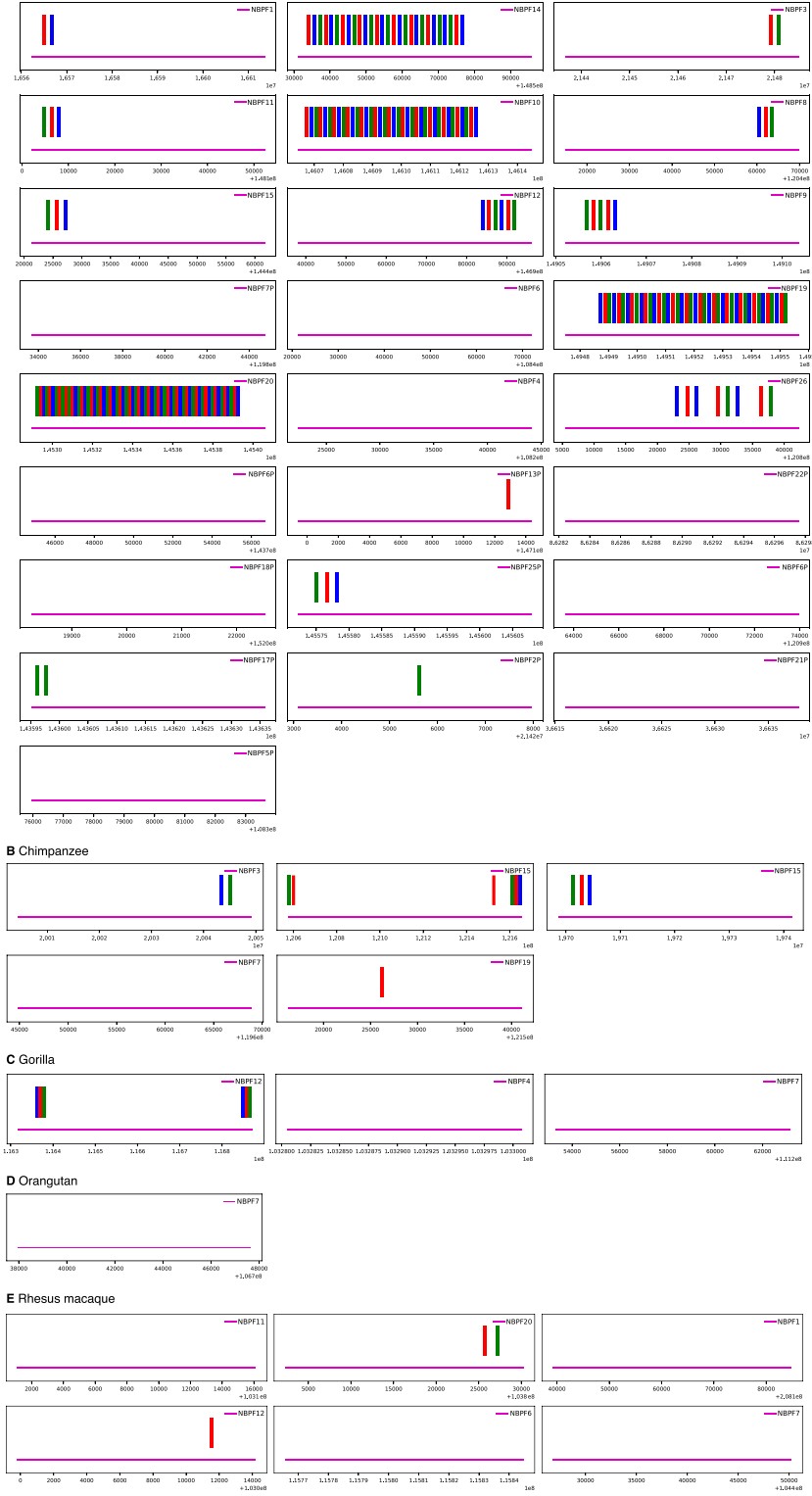

**Figure 3. NBPF monomer distribution along the National Center for Biotechnology Information (2022) genome assemblies.**
**(A)** Human, NC_000001.11. **(B)** Chimpanzee, NC_036879.1. **(C)** Gorilla, NC_044602.1. **(D)** Orangutan, NC_036903.1. **(E)** Rhesus macaque, NC_041754.1. Horizontally displayed is position along genome sequence within each NBPF gene. Each NBPF monomer is shown by a short vertical line: m1 (green), m2 (red), and m3 (blue). The canonical NBPF 3mer copies in tandem are present in four human NBPF genes: NBPF20, NBPF19, NBPF10, and NBPF14.

**Table 1.** Comparison of the human reference genome GRCh38 copy number and structural organization for an illustrative segment of tandemly organized NBPF higher order repeat copies to the result for 20 randomly chosen individual human genomes from the International Genome Sample Resource (the 1000 Genomes Project: http://www.internationalgenome.org/data-portal).

| Reference genome GRCh38 | | | | 20 individual genomes from Table S3 | | |
|---|---|---|---|---|---|---|
| Mon. No. | L(bp) | Mon. type | Div. (%) | L(bp) | Mon. type | Div. (%) |
| 67 | 1,544 | m03 | 1,78 | 1,544 | m03 | 1,78 |
| 68 | 1,628 | m02 | 0,44 | 1,628 | m02 | 0,44 |
| 69 | 1,584 | m01 | 1,56 | 1,584 | m01 | 1,56 |
| 70 | 1,542 | m03 | 1,29 | 1,542 | m03 | 1,29 |
| 71 | 1,625 | m02 | 1,78 | 1,544 | m02 | 1,78 |
| 72 | 1,587 | m01 | 1,78 | 1,544 | m01 | 1,78 |
| 73 | 1,542 | m03 | 1,84 | 1,542 | m03 | 1,84 |
| 74 | 1,629 | m02 | 2,27 | 1,629 | m02 | 2,27 |
| 75 | 1,599 | m01 | 1,62 | 1,599 | m01 | 1,62 |
| 76 | 1,542 | m03 | 2,09 | 1,542 | m03 | 2,09 |
| 77 | 1,631 | m02 | 1,45 | 1,631 | m02 | 1,45 |
| 78 | 1,591 | m01 | 1,62 | 1,591 | m01 | 1,62 |
| 79 | 1,542 | m03 | 1,72 | 1,542 | m03 | 1,72 |
| 80 | 1,622 | m02 | 1,33 | 1,622 | m02 | 1,33 |
| 81 | 1,584 | m01 | 1,49 | 1,584 | m01 | 1,49 |
| 82 | 1,541 | m03 | 1,78 | 1,541 | m03 | 1,78 |
| 83 | 1,628 | m02 | 0,76 | 1,628 | m02 | 0,76 |
| 84 | 1,584 | m01 | 1,49 | 1,584 | m01 | 1,49 |
| 85 | 1,542 | m03 | 1,84 | 1,542 | m03 | 1,84 |
| 86 | 1,629 | m02 | 1,20 | 1,629 | m02 | 1,20 |
| 87 | 1,589 | m01 | 1,62 | 1,589 | m01 | 1,62 |

Mon. No., ordinal number of NBPF monomer/Olduvai domain in global repeat map (GRM) analysis; L(bp), length of monomer; Mon. type, monomer type from GRM analysis; Div. (%), divergence of monomer with respect to the GRM consensus of the corresponding type. This illustrative segment presents the subsequence of monomers No. 67–87, which belong to tandemly organized canonical 3mer higher order repeat copies. Start position of the monomer no. 67 in the reference genome GRCh838 is 144424115.

indicates that Olduvai triplet proteins are the ultimate functional unit for Olduvai in humans.

## Discussion

Results of both HLS-searching and HOR-searching methods are largely congruent, in spite of using computational procedures with different divergence range. In the HOR copy searching procedure divergence range is much lower than in the HLS domain searching. On the other hand, the HLS-searching method provides in the first step also domains conserved among primates (CON), which can be obtained in the HOR-searching method only in the second step, after determining 3mer HORs and their repeat units.

From analysis of divergence in HOR-searching method by using GRM algorithm, the identified NBPF HOR copies have a pattern comparable with classical Willard's *α* satellite HORs in the centromeric regions (Warburton & Willard, 1996). For example, the average divergence among canonical HOR copies in HOR array

within the NBPF20 gene is ~1%, whereas among monomers within each HOR copy divergence is sizable, ~17–19%. Results obtained for NBPF monomer repeats and 3mer HOR copies by using GRM method are similar to the corresponding Olduvai domains and Olduvai triplets, respectively, obtained by using HLS searching method, with slight difference due to different range of divergences. Because of smaller divergence between repeats detected in the first step of analyzing a given genomic sequence, it is simpler to use the HOR-searching than the domain (monomer) searching method, but the final results for NBPF HORs/Olduvai triplets are nearly the same.

As pointed out in Reference Andries et al (2012), before the discovery of NBPF 3mer HORs in Reference Paar et al (2011) by using the GRM HOR-searching method, it was not noted that diverging HLS domains, now called Olduvai domains, are of three types, and forming a remarkable 3mer HOR pattern. The main advantage of the GRM HOR-searching algorithm is the characteristic small divergence between HOR copies. After initial discovery of NBPF 3mer HORs (Paar et al, 2011), the alternative way to identify HORs in a given sequence was the HLS domains searching in the first step, and then, by analyzing their mutual divergences, in the second step

to identify their triplets (i.e., Olduvai triplets/HOR copies) with small mutual divergence among triplets (O'Bleness et al, 2012; O'Bleness et al, 2014; Astling et al, 2017). In comparison of results obtained by these two methods, it should be taken into account that different terminology is used for the same pattern: DUF1220 HLS domains (now Olduvai domains) instead of NBPF monomer and HLS DUF1220 triplet (now Olduvai triplets) instead of NBPF 3mer HOR copy. It is noted that the HOR terminology was established and widely used in the pioneering work of Willard and co-workers for α satellite HORs (Warburton & Willard, 1996).

Using the HLS searching method, it was discovered that the ~1.6 kb repeat units in primate NBPF genes are human specific; their copy number gradually increases from nonhuman primate genomes to human genome: rhesus macaque 74 → common marmoset 75 → olive baboon 75 → gorilla 97 → orangutan 101 → chimpanzee 138 → human 302 (Fortna et al, 2004; Vandepoele et al, 2005; Popesco et al, 2006; Dumas & Sikela, 2009; Dumas et al, 2012; Zimmer & Montgomery, 2015). Here we point out to even more dramatical human exclusive discontinuous jump of the number of tandemly organized NBPF canonical 3mer HOR copies (Olduvai triplets) from 0 in nonhuman primates to 50 in the evolutionary step from chimpanzee to human genome. One might hypothesize whether this nonlinearity might involve some additional coherence in human specific brain evolution, beyond the merely summation of individual ~1.6-kb Olduvai domain contributions.

Additional two pronounced HORs in human chromosome 1 are the quartic-order 2mer (2mer [9mer]) HOR based on 39-bp primary hornerin repeat unit in hornerin gene and the α satellite 11mer HOR in the centromere region (Warburton & Willard, 1996; Paar et al, 2011). A study of more recent chimpanzee reference genome has revealed the presence of tandemly organized hornerin quartic order HORs in more complete sequence of chimpanzee chromosome 1, that is, that tandemly organized quartic order hornerin HOR copies are present also in chimpanzee, thus excluding the existence of exclusively human tandemly organized hornerin quartic copies (Romero et al, 2018).

On the other hand, the problem of chimpanzee α satellite 11mer HOR remains open because of still large unsequenced gaps (Table S5) in the current reference genome NC_036879.1 from The National Center for Biotechnology Information (NCBI). Thus, the only proven exclusively human tandemly organized HOR copies in primates' chromosome 1 are the NBPF 3mer HOR/Olduvai triplet copies.

Extensive studies of evolution of Olduvai triplets in *NBPF* genes have been performed using Olduvai method. Thanks to rather complete sequencing of *NBPF* genes, it was possible to identify Olduvai triplets along the evolutionary chain. It was shown that the number of Olduvai triplets gradually expands along the evolutionary chain to human genome. Because of equivalence of Olduvai triplets and NBPF 3mer HOR copies, the same results hold for the HOR-searching method.

The evolutionary problem is more difficult for α satellite 11mer HOR. GRM of human genome assembly NC_000001.11 shows a pronounced peak for α satellite 11mer HOR, which is absent in current genome assemblies of nonhuman primates chimpanzee (NC_036879.1), gorilla (NC_044602.1), orangutan (NC_036903.1) and Rhesus macaque (NC_041754.1) (Fig S1). This is obviously due to large gaps in available current sequencing of centromeric region of nonhuman genomes. Therefore, it is not possible at present to study evolution of 11mer HOR in primates.

## Materials and Methods

NBPF HORs are identified in NCBI assembly (2022) of human and nonhuman primate genomes (https://www.ncbi.nlm.nih.gov/genome/), using GRM algorithm. GRM algorithm is efficient and robust method convenient to identify and study in a given genomic sequence very large repeat units, like HORs (Paar et al, 2011; Gluncic & Paar, 2013; Gluncic et al, 2019; Vlahović et al, 2020). The computational noise in detecting repeats increases with increased length and/or complexity of HOR repeat unit, which can mask some peaks corresponding to HOR copies. Such background noise is significantly reduced by using GRM algorithm. The novelty of GRM approach is a direct mapping of symbolic DNA sequence into frequency domain using complete K-string ensemble instead of statistically adjusted individual K-strings optimized locally. This provides a straightforward identification of DNA repeats in frequency domain but avoids mapping of symbolic DNA sequence to numerical sequence, and uses K-string matching but avoids statistical methods and locally optimizing individual K-strings (Paar et al, 2011; Gluncic & Paar, 2013). In this way, characteristics of GRM algorithm are robustness with respect to deviations from ideal repeats (substitutions, insertions, deletions), straightforward and parameter-free identification of repeats, applicability to very large repeat units—both primary repeats and HORs, and straightforward determination of consensus lengths and consensus sequences for primary repeats and HORs. The GRM method is a straightforward method to provide a global repeat map in a GRM diagram, identifying all pronounced repeats in a given sequence, without any prior knowledge on repeats. In addition, once the consensus repeat unit is determined using GRM, in the next step it can be combined with search for dispersed HOR copies or individual constituting monomers.

### Conclusions

Using robust GRM algorithm, here the tandem repeats of NBPF HOR copies are searched in NCBI (2022) genome assemblies for nonhuman primates, chimpanzee, gorilla, orangutan, and Rhesus macaque, in comparison with human genome. The number of tandemly organized canonical NBPF HOR copies in human genome is 50, whereas in nonhuman primates is 0. It is hypothesized that the impact of NBPF monomners in human brain evolution is enhanced as a coherent effect when monomers are organized in tandem of HORs. This exclusively human specific NBPF HOR symmetry could involve a link in human cognitive development.

## Supplementary Information

# Acknowledgements

This work was supported by the QuantiXLie Centre of Excellence, a project cofinanced by the Croatian Government and European Union through the European Regional Development Fund—the Competitiveness and Cohesion Operational Programme (Grant KK.01.1.1.01.0004), and the grant IP-2019-04-2757 from Croatian Science Foundation.

## Author Contributions

M Glunčić: conceptualization, resources, data curation, software, formal analysis, supervision, funding acquisition, validation, investigation, visualization, methodology, project administration, and writing—original draft, review, and editing.
I Vlahović: data curation, software, and formal analysis.
M Rosandić: data curation, investigation, and visualization.
V Paar: conceptualization, investigation, visualization, methodology, and writing—original draft, review, and editing.

## Conflict of Interest Statement

The authors declare that they have no conflict of interest.

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
