## [Reviewer comments · Life Science Alliance]

Life Science Alliance

Tandemly repeated NBPF HOR copies (Olduvai triplets) - possible impact on human brain evolution

Matko Gluncic, Ines Vlahovic, Marija Rosandic, and Vladimir Paar

DOI: <https://doi.org/10.26508/lsa.202201306>

Corresponding author(s): Matko Gluncic, University of Zagreb

Review Timeline:

Submission Date:	2021-11-19
Editorial Decision:	2022-01-03
Revision Received:	2022-08-25
Editorial Decision:	2022-09-23
Revision Received:	2022-09-29
Accepted:	2022-09-30

Scientific Editor: Novella Guidi

Transaction Report:

January 3, 2022

Re: Life Science Alliance manuscript #LSA-2021-01306-T

Dr Matko Gluncic
Faculty of Science, University of Zagreb
Department of Physics
Bijenicka cesta 32
Zagreb 10000
Croatia

Dear Dr. Gluncic,

Thank you for submitting your manuscript entitled "Tandemly repeated NBPF HOR copies in non- human primates versus human genome - possible impact on human brain evolution" to Life Science Alliance. The manuscript was assessed by expert reviewers, whose comments are appended to this letter. We, thus, encourage you to submit a revised version of the manuscript back to LSA that responds to all of the reviewers' points.

Thank you for this interesting contribution to Life Science Alliance. We are looking forward to receiving your revised manuscript.

Sincerely,

B. MANUSCRIPT ORGANIZATION AND FORMATTING:

Reviewer #1 (Comments to the Authors (Required)):

Major comments: The authors use a unique bioinformatics technique termed the Global Repeat Map (GRM) algorithm to re-evaluate previously found higher order repeat (HOR) copies (also termed DUF1220 triplets and Olduvai triplets) within NBPF using updated genomic assemblies for human and non-human primates. Results show peaks at 1.6kb, 3.2kb and 4.8kb, which correspond to 1mer, 2mer, and 3mer copies, respectively. These results confirm that previously found NBPF HOR copies are found only in the updated human genome assembly and not in the updated non-human primate assemblies (chimpanzee, gorilla, orangutan, rhesus macaque). In total 50 NBPF HOR copies are identified in the human genome assembly. These findings confirm and update this groups previous findings (2011), and compliment other groups' findings that show a unique highly repetitive, human-specific amplification for NBPF domains, which may have contributed to human brain evolution. It would be helpful, space permitted, to have a more thorough discussion of evolutionary context of these findings and how they fit in with findings by other groups.

Minor comments: On a more minor point the authors state that the HOR terminology should be used as opposed to the HLS terminology, however the HLS/CON subtype terminology is informative and useful for distinguishing between the human specific HOR copies (HLS) and the domains conserved among primates (CON), both of which are found within human NBPF genes and have independent associations with brain phenotypes. Additionally, authors make no mention of the formal name change of these domains from DUF1220 to Olduvai (Sikela & van Roy, 2017). The text within Figure 2 is fairly small and difficult to read.

Reviewer #2 (Comments to the Authors (Required)):

In this paper the authors investigated the tandemly organization of NBPF HORs in the current human and nonhuman primate genome assemblies NCBI (2020), using the Global Repeat Map(GRM) algorithm. The main result is: The number of tandemly organized canonical NBPF HOR copies in human genome is 50, while in nonhuman primates is absent.

The paper is quite confusing because reading the title as well as the abstract and the introduction it is expected to find more strong results and an appropriate discussion about "contributor to the human brain evolution" is missing. -Since, the discussion paragraph starts by stressing the main advantage of using the GRM HOR-searching algorithm with respect to the monomer searching method; continuing with "The results of both methods are mutually similar for the study of NBPF higher order repeats". The same paragraph, rightly, pointed that the different terminology present in the literature have the same meaning: "DUF1220 HLS instead of NBPF monomer and HLS DUF1220 triplet instead of NBPF HOR copy. They conclude, based on alpha satellite DNA structure: "it seems more appropriate to use the HOR terminology also in the case of NBPF repeats". Therefore, a proper discussion on the effects of NBPF tandem repeats in brain evolution was not included. -In the same way, the conclusion paragraph just reports a statement: "Based on this results the author hypothesized that the impact of NBPF monomers in human brain evolution is dramatically enhanced when they are organized as tandem of HORs".

Overall, there are no descriptions about NBPF genes; intragenic loci where repeats are present; why and how these repeats should alterate NBPF genes etc..

I suppose the paper should be reorganized and rewritten adding also information such as: what is the relation between alpha satellite and NBPF repeat (or DUF1220 repeat)? are the dispersed euchromatic form of NBPF repeats comparable to dispersed form of a satellite DNA? if yes, why discuss only about the terminology and not about satellite DNA evolution and the mechanism of dispersion? why hypothesized that the putative influence of repeats occurs only when they are organized in HOR and not when they are present in the form of monomer? do the authors know the influence of dispersed form of satellite DNA monomer on gene expression? etc..

Reviewer #3 (Comments to the Authors (Required)):

In this paper the Authors investigated whether the tandemly organized canonical NBPF 3mer HORs are present in current genome assembly in both human and nonhuman primate genomes.

I think the paper is very poor and does not represent a major scientific advancement in the field. In addition, I think it needs to be partially rewritten. The introduction is very confusing. While studying it at length I found it "cumbersome" and complicated to follow, also, without a picture it is hard to imagine the HORs the authors are talking about.

The main problem is in the conclusions. The authors, by finding a particular organization on chromosome 1 of Homo sapiens that is missing in other primates, conclude that this genomic structure plays a key role in the development of cognitive abilities that differentiate humans from other primates.

I might well agree, the hypothesis is compelling. However, it does not take into account the diversity among different humans (and among other primates). The authors have reached their conclusions by analyzing a single human sequence. The paper would be greatly improved if the Authors could also use their method on other genetic sequences (found in the literature) in order to observe which is the diversity (in terms of copy number and structural organization) of this genomic portion.

Finally, the Discussion is really poor and the genomic coordinates indicated at page 9 are really bad organised.

Rebuttal letter

„Tandemly repeated NBPF HOR copies in non-human primates versus human genome - possible impact on human brain evolution“

We thank referees for their insightful comments. Closely following their very useful comments, requests, and recommendations we have made the following changes to the original article. All changes in the manuscript and rebuttal letter are marked in blue.

Revised title

In accordance with suggestion by reviewer to comment on the novel name Olduvai, introduced recently in the literature, we propose a slight extension of the title by adding “(Olduvai triplets)”. In accordance with journal’s 100 character limit (including spaces) we have shortened the title by deletion “in non-human primates versus human genome”. New title is:

Tandemly repeated NBPF HOR copies (Olduvai triplets) - possible impact on human brain evolution

Revised abstract

In accordance with suggestion of reviewers that the paper should be reorganized and rewritten, adding also additional information, we have revised the Abstract as follows:

Abstract

Previously it was found that the Neuroblastoma Breakpoint Family (NBPF) gene repeat units of ~1.6 kb, have an important role in human brain evolution and function. The higher order organization of these repeat units has been discovered by both methods, the HOR-searching method and the HLS searching method. Using the HOR searching method with Global Repeat Map algorithm (GRM), here we identified the tandemly organized NBPF HORs (Higher Order Repeats) in the human and nonhuman primate NCBI reference genomes. We identified 50 tandemly organized canonical 3mer NBPF HOR copies (Olduvai triplets), but none in nonhuman primates chimpanzee, gorilla, orangutan and Rhesus macaque. This discontinuous jump in tandemly organized HOR copy number is in sharp contrast to the known gradual increase of the number of Olduvai domains (NBPF monomers) from nonhuman primates to human, especially from ~138 in chimpanzee to ~300 in human genome. Using the same GRM method we have also determined the 3mer tandems of canonical 3mer HOR copies in 20 randomly chosen human genomes (10 male and 10 female). In all cases we found the same 3mer HOR copy numbers as in

the case of the reference human genome, with no mutation. On the other hand, some point mutations with respect to reference genome are found for some NBPF monomers which are not tandemly organized in canonical HORs.

Revised Introduction

Reviewer 1

“On a more minor point the authors state that the HOR terminology should be used as opposed to the HLS terminology, however HLS/CON subtype terminology is informative and useful for distinguishing between the human specific HOR copies (HLS) and the domains conserved among primates (CON), both of which are found within human NBPF genes and have independent associations with brain phenotypes.

Additionally, authors make no mention of the formal name change of these domains from DUF1220 to Olduvai (Sikela & van Roy, 2017). “

Reviewer 2

“Since the discussion paragraph starts by stressing the main advantage of using the GRM HOR-searching algorithm with respect to the monomer searching method, continuing with “The results of both methods are mutually similar for both the study of NBPF higher order repeats.” The same paragraph, rightly, pointed that the different terminology present in the literature have the same meaning: “DUF1220 HLS instead of NBPF HOR copy. They conclude, based on alpha DNA structure: “it seems more appropriate to use the HOR terminology also in the case of NBPF repeats.

A proper discussion of NBPF tandem repeats in brain evolution was not included. In the same way, the conclusion paragraph just reports the statement: “Based on this results the author hypothesized that the impact of NBPF monomers in human brain evolution is dramatically enhanced when they are organized as tandem HORs”.

Overall there are no descriptions about NBPF genes.

I suppose the paper should be reorganized and rewritten adding also information such as: what is the relation between alpha satellite and NBPF repeat (or DUF1220 repeat)?”

Reviewer 3

“I think the paper needs to be partially rewritten. The introduction is very confusing. While studying it at length I found it “cumbersome” and complicated to follow, also, without a picture it is hard to imagine the HORs the authors are talking about.”

Closely following these very useful comments, requests, and recommendations by the reviewers we have rewritten the Introduction as follows.

The ~1.6 kb repeat units in the Neuroblastoma BreakPoint Family (NBPF) genes

The Neuroblastoma BreakPoint Family (NBPF) genes in human chromosome 1 contain a pronounced repetitive structure of ~1.6 kb repeat units (Fortna et al. 2004; Vandepoele et al. 2005; Popesco et al. 2006), with sizable divergence (~20%) among neighboring repeat units. These repeat units encode the protein domain previously named DUF1220 domain (Vandepoele et al. 2005), which was changed to Olduvai domain (Sikela and van Roy 2017). These repeat units within NBPF genes were previously also called NBPF repeats or NBPF monomers (Paar et al. 2011) (Fig. 1), in accordance with repeat terminology used by (Willard 1985) in studies of alpha satellite repeats. Each of Olduvai domains (NBPF repeat units) is characterized by a structure with two evenly spaced exons (Popesco et al. 2006; Paar et al. 2011). Comprehensive studies have shown that the Olduvai domain copy number was correlated with brain size, cortical neuron number, IQ scores, cognitive aptitude, evolution and with brain pathologies (autism, schizophrenia, microcephaly, macrocephaly, neuroblastoma) (Vandepoele et al. 2005; Popesco et al. 2006; Vandepoele et al. 2008; Andries et al. 2012; Dumas et al. 2012; Davis et al. 2014; Keeney et al. 2014; Quick et al. 2016; Astling et al. 2017; Mitchell and Silver 2018; Fiddes et al. 2019; Heft et al. 2020).

HLS-searching method

Comparing sequences of repeated Olduvai domains, it was found that they are predominantly of three types, which were referred to as HLS-1, HLS-2 and HLS-3, and most often they appear in triplets (O'Bleness et al. 2012; O'Bleness et al. 2014) (Fig. 1). Due to recent change of terminology, which was fully justified (Sikela and van Roy 2017), these HLS triplets were referred to as Olduvai triplets. HLS searching method identifies Olduvai domains in a given genomic sequence in the first step and compares their divergence to determine Olduvai triplets in the second step.

HOR-searching method

Centromeric regions of primate chromosomes are largely built from alpha satellite repeat units of length ~171 bp, which have been extensively studied (Manuelidis 1978; Willard 1985; Jorgensen et al. 1987; Tyler-Smith and Brown 1987; Waye and Willard 1987; Choo et al. 1991; Jurka et al. 1996; Warburton and Willard 1996; Alexandrov et al. 2001; Rosandic et al. 2003; Rudd and Willard 2004; Jurka et al. 2005; Paar et al. 2005; Warburton et al. 2008; Aldrup-Macdonald and Sullivan 2014; Miga 2017; Sullivan et al. 2017; Lower et al. 2018; Gluncic et al. 2019; Uralsky et al. 2019). Neighboring alpha satellite monomers (primary repeat units) diverge sizably from each other (~20-40%), but some stretches of n monomers are often organized into secondary repeat n mer HOR copies, with small mutual divergence between HOR copies (less than 5%, in some cases even below 1%). Thus, divergence between HOR copies is much smaller than between monomers within each HOR copy (Warburton and Willard 1996). Pronounced alpha satellite n mer HORs in human genome are chromosome specific, for example: 2mer and 11mer in chromosome

1; 15mer in chromosome 8, 5mer in chromosome 11, 12mer in chromosome X, 34mer in chromosome Y (Alexandrov et al. 2001).

An efficient and robust computational tool to identify large HOR copies, as for example alpha satellite HORs, is the Global Repeat Map (GRM) algorithm (Paar et al. 2011; Gluncic and Paar 2013; Gluncic et al. 2019; Vlahović et al. 2020). HOR method to determine alpha satellite repeat structure is based on identifying alpha satellite HOR copies in a given sequence in the first step, and in the second step deducing alpha satellite monomers from the HOR structure. Thus, HOR method enables identification of HOR copies without any prior knowledge of the primary repeats on which the HORs are based. Because the divergence among HOR copies is much smaller than the divergence among its underlying primary repeats, the scope of computations is simpler in the case of using HOR method than in the method of using monomer identification.

GRM algorithm identifies all pronounced HORs in a given genomic sequence. For example, applying in 2011 the GRM algorithm to the Builds 36.3 assembly of human chromosome 1 we obtained three HORs (Paar et al. 2011):

- 1) the already known alpha satellite 11mer HOR (based on ~171 bp primary alpha satellite repeat unit);
- 2) the novel 3mer HOR, based on previously known ~1.6 kb primary repeat units from NBPF genes in human chromosome 1;
- 3) the novel quartic HOR, based on known 39 bp hornerin primary repeat unit in human chromosome 1.

While the 11mer alpha satellite HORs are located in the centromere, the last two HORs are located within NBPF genes and Hornerin genes, respectively.

We also note that by applying GRM algorithm to genomic sequences of insects, the HOR pattern (based on T-cast-360 monomers of 331, 361, 362, and 369 bp primary repeat units), has been also discovered in beetle *Tribolium castaneum* (Vlahovic et al. 2017), which is evolutionary very distant from primates.

Using the robust GRM algorithm, we have first discovered in Build 36.3 assembly of human chromosome 1 the ~4,770 bp peak of 3mer HOR (Paar et al. 2011). The GRM diagram for this ~4,770 bp consensus sequence revealed two additional internal GRM peaks, at ~1.6 kb and ~3.2 kb. The GRM diagram for the 3.2-kb peak also shows that its consensus sequence consists of two ~1.6 kb monomer repeats. Altogether, the ~4,770 bp consensus sequence consists of a tandem of three ~1.6 kb monomer repeats of consensus lengths 1,623, 1,593 and 1,554 bp, respectively. The corresponding consensus monomers were denoted m01, m02, and m03, respectively. In this way, we discovered that the ~4,770 bp repeat copies are the 3mer higher order repeats (HORs). Since these 3mer HORs are located within the NBPF genes in human chromosome 1, we named them NBPF 3mer HORs, and their constituent ~1.6 kb monomers were referred to as NBPF monomers in accordance with Willard's terminology for HORs (Warburton and Willard 1996). Each consensus monomer consists of two exons and three introns. Each consensus monomer contains in succession intron-exon-intron-exon-intron. In the 1,554 bp consensus monomer their lengths amount to ~ 8%, ~4%, ~39%, ~12% and ~37 % of monomer length. In the 1,593 bp consensus monomer the length of second intron increases by ~8%, and in the 1,623 bp consensus monomer additionally the length of the first intron increases by ~6%.

In the Build 36.3 genomic assembly of human chromosome1, we identified 47 HOR copies forming tandems, but only 34 of these 47 were canonical as shown in Fig. 8 from Ref. (Paar et al. 2011). The three constituent monomers of ~1.6 kb correspond to HLS DUF1220 domains of (Fortna et al. 2004; Vandepoele et al. 2005; Popesco et al. 2006). As pointed out in (Andries et al. 2012), before the discovery of NBPF 3mer HOR by (Paar et al. 2011), it was not realized by HLS-searching method that the HLS domains form the triplet organization, i.e., that the diverging DUF1220 HLS domains are of three types, forming a remarkable triplet organization – the higher order repeats. On the other hand, the Build 2.1 (2010) genomic assembly of chimpanzee chromosome 1 did not show tandemly organized NBPF HOR copies, although 14 individual NBPF HOR copies (not organized into tandem) were identified and the total number of dispersed NBPF monomers was 48 (Paar et al. 2011). Also, tandemly organized NBPF HOR copies were not found in genomic assembly WUSTL Pongo_albelii-2.0.2 (2010) of orangutan and in Build 1.1 (2010) of Rhesus macaque (Paar et al. 2011). On this basis we hypothesized that the NBPF HOR tandem repeats (47 NBPF HOR copies in human versus 0 in nonhuman primate genome) reveal the NBPF HOR tandem repeats are human specific, possibly contributing to human brain evolution and human-chimpanzee divergence (Paar et al. 2011).

A possible challenge to such hypothesis has arisen from another complex HOR pattern, discovered by GRM algorithm within the hornerin (HRNR) gene in Build 36.3 genome assembly of human chromosome 1 (Paar et al. 2011): nine 39 bp primary repeat HRNR units organized into ~0.35 kb secondary repeat units → two ~0.35 kb secondary HOR repeat units organized into ~0.7 kb tertiary HOR repeat units → and finally two 0.7 kb tertiary HOR repeat units organized into ~1.4 kb quartic HOR repeat units. In this, the 1,410 bp repeat unit represents the quartic HOR repeat unit, which is tandemly repeated, as the most complex multi-step HOR pattern discovered so far. In Ref. (Paar et al. 2011) the quartic HOR was detected in the human genome only, and no quartic HOR counterpart was found in then-available sequenced chimpanzee genome.

The more recent investigation by (Romero et al. 2018) of HRNR from more complete NCBI human genomic assembly NC 000001.11 (2018) fully confirmed the HRNR quartic HOR formation described by (Paar et al. 2011). However, (Romero et al. 2018) found that the tandem quartic HOR formation of HRNR was conserved also in more recent reference genomes of chimpanzee and other nonhuman primates, except the crab-eating macaque. This showed that the tandemly organized HRNR quartic HOR copies are not exclusively human-specific, as appeared for reference genome in 2011, but are also present in more recent NCBI reference genome sequences of most primates (Romero et al. 2018).

Prompted by that development, here we investigate whether the tandemly organized canonical NBPF 3mer HORs are present in current genome assemblies in both human and nonhuman primate genomes, or are exclusively human-specific, as was indicated by previous chimpanzee reference genome.

Reviewer 3 “I think the paper needs to be partially rewritten. The introduction is very confusing. While studying it at length I found it “cumbersome” and complicated to follow, also, without a picture it is hard to imagine the HORs the authors are talking about.”

Closely following this suggestion by Reviewer 3 we have added the proposed picture.

Figure 1. HLS vs. HOR terminology.

Revised Results

Reviewer 3 “The main problem is in the conclusions. The authors, by finding a particular organization on chromosome 1 of *Homo sapiens* that is missing in other primates, conclude that this genomic structure plays a key role in the development of cognitive abilities that differentiate humans from other primates. I might well agree, the hypothesis is compelling. However, it does not take into account the diversity among different humans (and among other primates). The authors have reached their conclusions by analyzing a single human sequence. The paper would be greatly improved if the Authors could also use their method on other genetic sequences (found in the literature) in order to observe which is the diversity (in terms of copy number and structural organization) of this genomic portion.”

Closely following this recommendation by the reviewer, we have extended our HOR study to additional human genetic sequences (20 randomly chosen individuals from the literature) which

is included in the following additional subsection in the Results section and additional Tables (additional Table 1 and additional Supplementary Table 3)

Comparison of copy number and structural organization of tandemly organized NBPF 3mer HOR copies of 20 randomly chosen individual human genomes from The International Genome Sample Resource (IGSR) and human reference genome GRCh38

In order to observe which is the diversity of this genomic portion in terms of copy number and mutations in different genomes, we have applied our HOR-finding method to sequences of 20 randomly chosen individuals from The International Genome Sample Resource (IGSR) (The 1000 Genomes Project: <https://www.internationalgenome.org/data-portal>) (Supplementary Table 3). In all 20 genomes we obtained exactly the same copy numbers and the same consensus HOR sequences (with no mutations) equal to results for reference genome GRCh38, as illustrated by Table 1. (For full results on all 20 genomes see Supplementary material 2 Table, where monomers marked by asterisk do not belong to canonical 3mer HOR copies and some have a point mutation with respect to results for reference genome GRCh38.) In the table the same value of divergence for the corresponding monomers in canonical HOR copies for all 20 individual genomes and reference genome means that monomers in canonical HOR copies in all 20 individual genomes have no mutation with respect to the reference genome. On the other hand, a sizable number of monomers from HOR copies outside of tandemly organized 3mer HOR copies have some point mutations with respect to the reference genome, although their HOR copy number and structural organization (monomer types) are the same as for reference genome. For example, the every monomer No. 30 in 20 individual genomes from Suppl. Table S3 has the same monomer length 1529 bp and monomer type m01, but only 8 of them have the same divergence value (14.54 %) as the corresponding monomer No. 30 for reference genome (i.e., no point mutation with respect to the reference genome), while the remaining 12 individual genomes have slightly different divergences (14.60 %, 14.67 %) (i.e., point mutation with respect to the reference genome). Due to large length of NBPF monomer (~1.6 kb) such rare point mutations cannot influence the NBPF HOR structure and copy numbers, but are restricted only to slight difference of monomer sequences. In this sense, the tandemly repeated 3mer consensus HOR pattern may be considered as a kind of characteristic signature of human genome.

Additional Table 1

Table 1. Comparison of the human reference genome GRCh38 copy number and structural organization for an illustrative segment of tandemly organized NBPF HOR copies to the result for 20 randomly chosen individual human genomes from The International Genome Sample Resource (IGSR) (The 1000 Genomes Project: <https://www.internationalgenome.org/data-portal>). Mon. No., ordinal number of NBPF monomer/Olduvai domain in GRM analysis; L(bp), length of monomer; Mon. type, monomer type from GRM analysis; Div. (%), divergence of monomer with respect to the GRM consensus of the corresponding type. This illustrative segment presents the subsequence of monomers No. 67-87, which belong to tandemly organized canonical 3mer HOR copies. Start position of the monomer No. 67 in the reference genome GRCh838 is 144424115.

Reference genome GRCh38				20 individual genomes from Supp. Table 3		
Mon. No.	L(bp)	Mon.type	Div. (%)	L(bp)	Mon.type	Div. (%)
67	1544	m03	1,78	1544	m03	1,78
68	1628	m02	0,44	1628	m02	0,44
69	1584	m01	1,56	1584	m01	1,56
70	1542	m03	1,29	1542	m03	1,29
71	1625	m02	1,78	1544	m02	1,78
72	1587	m01	1,78	1544	m01	1,78
73	1542	m03	1,84	1542	m03	1,84
74	1629	m02	2,27	1629	m02	2,27
75	1599	m01	1,62	1599	m01	1,62
76	1542	m03	2,09	1542	m03	2,09
77	1631	m02	1,45	1631	m02	1,45
78	1591	m01	1,62	1591	m01	1,62
79	1542	m03	1,72	1542	m03	1,72
80	1622	m02	1,33	1622	m02	1,33
81	1584	m01	1,49	1584	m01	1,49
82	1541	m03	1,78	1541	m03	1,78
83	1628	m02	0,76	1628	m02	0,76
84	1584	m01	1,49	1584	m01	1,49
85	1542	m03	1,84	1542	m03	1,84
86	1629	m02	1,20	1629	m02	1,20
87	1589	m01	1,62	1589	m01	1,62

Addition Supplementary Table 3

Supplementary Table 3. Twenty randomly chosen individual human genomes (10 male, 10 female) from The International Genome Sample Resource (IGSR) (The 1000 Genomes Project: <https://www.internationalgenome.org/data-portal>).

Sample	Seks	Populations	Biosample ID	Cell line source
HG00107	male	British in England and Scotland, European Ancestry	SAME123947	HG00107 at Coriell
HG00121	female	British in England and Scotland, European Ancestry	SAME122873	HG00121 at Coriell
HG00359	female	Finnish in Finland, European Ancestry	SAME125127	HG00359 at Coriell

HG00366	male	Finnish in Finland, European Ancestry	SAME124541	HG00366 at Coriell
HG00525	female	Han Chinese South, East Asian Ancestry	SAME123242	HG00525 at Coriell
HG00532	male	Han Chinese South, East Asian Ancestry	SAME125268	HG00532 at Coriell
HG00640	male	Puerto Rican in Puerto Rico, American Ancestry	SAME123434	HG00640 at Coriell
HG01162	female	Puerto Rican in Puerto Rico, American Ancestry	SAME1839707	HG01162 at Coriell
HG01345	female	Colombian in Medellin, Colombia, American Ancestry	SAME124863	HG01345 at Coriell
HG01491	male	Colombian in Medellin, Colombia, American Ancestry	SAME124140	HG01491 at Coriell
HG01509	male	Iberian populations in Spain, European Ancestry	SAME124940	HG01509 at Coriell
HG01605	female	Iberian populations in Spain, European Ancestry	SAME123459	HG01605 at Coriell
HG01794	female	Chinese Dai in Xishuangbanna, China, East Asian Ancestry	SAME124218	HG01794 at Coriell
HG00866	male	Chinese Dai in Xishuangbanna, China, East Asian Ancestry	SAME123549	HG00866 at Coriell
HG01597	female	Kinh in Ho Chi Minh City, Vietnam, East Asian Ancestry	SAME125226	HG01597 at Coriell
HG01864	male	Kinh in Ho Chi Minh City, Vietnam, East Asian Ancestry	SAME123197	HG01864 at Coriell
HG01888	male	African Caribbean in Barbados, African Ancestry	SAME122852	HG01888 at Coriell
HG01915	female	African Caribbean in Barbados, African Ancestry	SAME1839757	HG01915 at Coriell
HG01922	female	Peruvian in Lima, Peru, American Ancestry	SAME123787	HG01922 at Coriell
HG01934	male	Peruvian in Lima, Peru, American Ancestry	SAME123589	HG01934 at Coriell

Addition Supplementary Material 2 Table

In order to verify our results compressed in Table 1 we have included full results of our HOR-finding method to sequences of 20 randomly chosen individuals from The International Genome Sample Resource (IGSR) (The 1000 Genomes Project: <https://www.internationalgenome.org/data-portal>).

Revised Discussion

Reviewer 1

“It would be helpful, space permitted, to have a more thorough discussion of evolutionary context of these findings and how they fit in with findings by other groups.”

Reviewer 2

“Since the discussion paragraph starts by stressing the main advantage of using the GRM HOR-searching algorithm with respect to the monomer searching method, continuing with “The results

of both methods are mutually similar for both the study of NBPF higher order repeats.” The same paragraph, rightly, pointed that the different terminology present in the literature have the same meaning: “DUF1220 HLS instead of NBPF HOR copy. They conclude, based on alpha DNA structure: “it seems more appropriate to use the HOR terminology also in the case of NBPF repeats.

A proper discussion of NBPF tandem repeats in brain evolution was not included. In the same way, the conclusion paragraph just reports the statement: “Based on this results the author hypothesized that the impact of NBPF monomers in human brain evolution is dramatically enhanced when they are organized as tandem HORs”.

Overall there are no descriptions about NBPF genes“

“I suppose the paper should be reorganized and rewritten adding also information such as: what is the relation between alpha satellite and NBPF repeat (or DUF1220 repeat)?”

Following comments and recommendations by reviewers we have rewritten the section Discussion. The new text Discussion is as follows:

Results of both HLS-searching and HOR-searching methods are largely congruent, in spite of employing computational procedures with different divergence range. In the HOR copy searching procedure divergence range is much lower than in the HLS domain searching. On the other hand, the HLS-searching method provides in the first step also domains conserved among primates (CON), which can be obtained in the HOR-searching method only in the second step, after determining 3mer HORs and their repeat units.

From analysis of divergence in HOR-searching method by using GRM algorithm, the identified NBPF HOR copies have a pattern comparable to classical Willard’s alpha satellite HORs in the centromeric regions (Warburton and Willard 1996). For example, the average divergence among canonical HOR copies in HOR array within the NBPF20 gene is ~1%, while among monomers within each HOR copy divergence is sizable, ~17-19%. Results obtained for NBPF monomer repeats and 3mer HOR copies by using GRM method are similar to the corresponding Olduvai domains and Olduvai triplets, respectively, obtained by using HLS searching method, with slight difference due to different range of divergences. Because of smaller divergence between repeats detected in the first step of analyzing a given genomic sequence, it is simpler to use the HOR-searching than the domain (monomer) searching method, but the final results for NBPF HORs / Olduvai triplets are nearly the same.

As pointed out in Ref. (Andries et al. 2012), before the discovery of NBPF 3mer HORs in Ref. (Paar et al. 2011) by using GRM HOR-searching method, it was not noted that diverging HLS domains, now called Olduvai domains, are of three types, and forming a remarkable 3mer HOR pattern. The main advantage of the GRM HOR-searching algorithm is the characteristic small divergence between HOR copies. After initial discovery of NBPF 3mer HORs (Paar et al. 2011), the alternative way to identify HORs in a given sequence was the HLS domains searching in the first step, and then, by analysing their mutual divergences, in the second step to identify their triplets (i.e., Olduvai triplets/HOR copies) with small mutual divergence among triplets (O’Bleness et al. 2012; O’Bleness et al. 2014; Astling et al. 2017). In comparison of results obtained by these

two methods, it should be taken into account that different terminology is used for the same pattern: DUF1220 HLS domains (now Olduvai domains) instead of NBPF monomer and HLS DUF1220 triplet (now Olduvai triplets) instead of NBPF 3mer HOR copy. It is noted that the HOR terminology was established and widely used in the pioneering work of Willard and co-workers for alpha satellite higher order repeats (Warburton and Willard 1996).

Using the HLS searching method, it was discovered that the ~1.6 kb repeat units in primate NBPF genes are human specific; their copy number gradually increases from nonhuman primate genomes to human genome: rhesus macaque 74 → common marmoseth 75 → olive baboon 75 → gorilla 97 → orangutan 101 → chimpanzee 138 → human 302 (Fortna et al. 2004; Vandepoele et al. 2005; Popesco et al. 2006; Dumas and Sikela 2009; Dumas et al. 2012; Zimmer and Montgomery 2015). Here we point out to even more dramatical human exclusive discontinuous jump of the number of tandemly organized NBPF canonical 3mer HOR copies (Olduvai triplets) from 0 in nonhuman primates to 50 in the evolutionary step from chimpanzee to human genome. One might hypothesize whether this nonlinearity might involve some additional coherence in human specific brain evolution, beyond the merely summation of individual ~1.6 kb Olduvai domain contributions.

Additional two pronounced HORs in human chromosome 1 are the quartic-order 2mer (2mer (9mer)) HOR based on 39 bp primary hornerin repeat unit in hornerin gene and the alpha satellite 11mer HOR in the centromere region (Warburton and Willard 1996; Paar et al. 2011). A study of more recent chimpanzee reference genome has revealed the presence of tandemly organized hornerin quartic order HORs in more complete sequence of chimpanzee chromosome 1, i.e., that tandemly organized quartic order hornerin HOR copies are present also in chimpanzee, thus excluding the existence of exclusively human tandemly organized hornerin quartic copies (Romero et al. 2018).

On the other hand, the problem of chimpanzee alpha satellite 11mer HOR remains open due to still large unsequenced gaps (Supplementary Table 4) in the current reference genome NC_036879.1 from The National Center for Biotechnology Information (NCBI). Thus, the only proven exclusively human tandemly organized HOR copies in primates' chromosome 1 are the NBPF 3mer HOR / Olduvai triplet copies.

Extensive studies of evolution of Olduvai triplets in NBPF genes have been performed using Olduvai method. Thanks to rather complete sequencing of NBPF genes, it was possible to identify Olduvai triplets along the evolutionary chain. It was shown that the number of Olduvai triplets gradually expands along the evolutionary chain to human genome. Due to equivalence of Olduvai triplets and NBPF 3mer HOR copies, the same results hold for the HOR-searching method.

The evolutionary problem is more difficult for alpha satellite 11mer HOR. GRM of human genome assembly NC_000001.11 shows a pronounced peak for alpha satellite 11mer HOR, which is absent in current genome assemblies of nonhuman primates chimpanzee (NC_036879.1), gorilla (NC_044602.1), orangutan (NC_036903.1) and Rhesus macaque (NC_041754.1) (Supplementary Fig. S1). This is obviously due to large gaps in available current sequencing of centromeric region of nonhuman genomes. Therefore, it is not possible at present to study evolution of 11mer HOR in primates.

Supplementary Figure 1. GRM diagram for referent genome assemblies (NC_000001.11, NC_0368791.1, NC_044602.1, NC_036903.1 and NC_041754.1) of human and nonhuman primates. The human genome assembly has only small nonsequenced gaps in the centromeric region and therefore the peaks corresponding to alpha satellite *n*mers are seen: 11mer, 6mer, 4mer and 2mer. The longest HOR repeat copies in human chromosome 1 are 11mer. On the other hand, in nonhuman primates sizable segments of their genomes are are unsequenced, with largest nonsequenced gap od 20 Mb, so it is possible/probable that 11mer and other alpha satellite HORs lie in still unsequenced gaps and therefore are absent in GRM diagram for current chimpanzee and other nonhuman primates' reference genome sequences.

Supplementary Table 4. Large (> 1 kbp) unsequenced gaps in chimpanzee NCBI referent genome assembly NC_036879.1.

Positio		Length
start	end	(bp)
11.509.071	11.516.576	7.505
19.941.934	19.974.289	32.355
23.848.377	23.929.707	81.330
70.812.876	70.998.789	185.913
82.023.747	82.211.433	187.686
82.294.026	82.416.870	122.844
108.185.908	108.294.823	108.915
111.861.442	111.954.570	93.128
112.401.096	112.568.090	166.994
112.721.577	113.342.907	621.330
116.466.791	116.495.019	28.228
119.819.313	120.120.942	301.629
121.685.064	121.883.423	198.359
122.360.815	122.562.488	201.673
122.710.087	122.826.473	116.386
122.940.370	122.977.392	37.022
134.440.684	134.690.240	249.556

136.418.572	136.521.484	102.912
149.698.815	149.753.333	54.518
171.924.441	171.961.078	36.637
172.096.401	172.097.416	1.015
203.582.007	203.593.435	11.428
217.768.293	217.816.312	48.019
223.741.069	223.745.773	4.704

Revised Conclusions

Reviewer #2:

In the same way, the conclusion paragraph just reports a statement: "Based on this results the author hypothesized that the impact of NBPF monomers in human brain evolution is dramatically enhanced when they are organized as tandem of HORs".

Following comments and recommendations by reviewer we have changed the last sentence in section Conclusions: "It is hypothesized that the impact of NBPF monomers in human brain evolution is dramatically enhanced when they are organized as tandem of HORs.", into:

It is hypothesized that the impact of NBPF monomers in human brain evolution is enhanced as a coherent effect when monomers are organized in tandem of HORs. This exclusively human specific NBPF higher order repeat symmetry could involve a link in human cognitive development.

Technical improvements

Reviewer 2

"The text within Fig. 2 is fairly small and difficult to read."

In revised manuscript we have improved the presentation of Fig. 2 to become easier to read and have also moved the following part of the text below Figure 2 in Supplement:

Start positions of chimpanzee HOR copies

Start positions of Tandemly organized canonical 3mer NBPF HORs:

145292627, 145297386, 145308500, 145313256, 145318028, 145322790, 145327538, 145332304, 145337060, 145341811, 145346569, 145351341, 145356096, 145360849, 145365604, 145370357, 145375111, 145379863, 145384618, 146070571, 146075293, 146080029, 146084765, 146089486, 146094204, 146098926, 146103648, 146108357, 146113075, 146117794, 148537255, 148541969, 148546677, 148551385, 148556097, 148560831, 148565558, 149491881, 149496644, 149501399, 149506160, 149510921, 149515,677, 149520430, 149525187, 149529916, 149534667, 149539418, 149544155, 149548892.

Start positions of noncanonical 2mer variant NBPF HOR copies or appearing due to uncertainty of one monomer representation in a 3mer NBPF HOR consensus: 120460379, 120826179, 120832728, 144424115, 145389368, 145575066, 146122530, 146983938, 146988696, 148104784, 148573477, 149059942, 149487138.

Start positions of 3mer NBPf HOR with divergence > 5% of at least one constituent monomer with respect to consensus: 16565065, 21479230, 120823098, 145302139, 145305317, 146067427, 148534108, 148570288, 149056759.

A Human

B Chimpanzee

C Gorilla

D Orangutan

E Rhesus macaque

Revised References

In accordance with recommendations by reviewers, several additional references are included in the revised References:

Sikela JM, van Roy F. 2017. Changing the name of the NBPF/DUF1220 domain to the Olduvai domain. *F1000Res* **6**: 2185.

Zimmer F, Montgomery SH. 2015. Phylogenetic Analysis Supports a Link between DUF1220 Domain Number and Primate Brain Expansion. *Genome Biol Evol* **7**: 2083-2088.

September 23, 2022

RE: Life Science Alliance Manuscript #LSA-2021-01306-TR

Prof. Matko Gluncic
University of Zagreb
Faculty of Science
Bijenicka cesta 32
Zagreb 10000
Croatia

Dear Dr. Gluncic,

Thank you for submitting your revised manuscript entitled "Tandemly repeated NBPF HOR copies (Olduvai triplets) - possible impact on human brain evolution". We would be happy to publish your paper in Life Science Alliance pending final revisions necessary to meet our formatting guidelines.

- please address Reviewer 1's remaining comments
- please upload your manuscript text as an editable doc file
- please upload your main and supplementary figures as single files
- please upload your table files as editable doc or excel files or include them in the editable doc file of your manuscript
- please add ORCID ID for corresponding author-you should have received instructions on how to do so
- please add a summary blurb/alternate abstract and a category to our system
- please add the Twitter handle of your host institute/organization as well as your own or/and one of the authors in our system
- please incorporate the Supplemental Material "Start positions of chimpanzee HOR copies" section into the main manuscript
- Supplemental Material 2 should be provided as a Table with a legend
- please consult our manuscript preparation guidelines <https://www.life-science-alliance.org/manuscript-prep> and make sure your manuscript sections are in the correct order and add a separate section for your figure legends (main and supplementary figures and table legends)

A. FINAL FILES:

B. MANUSCRIPT ORGANIZATION AND FORMATTING:

Sincerely,

Reviewer #1 (Comments to the Authors (Required)):

Response to revised manuscript:

1. In the revised manuscript the authors have done a good job of including updated nomenclature, e.g. Olduvai domain, Olduvai triplet, etc.
2. The paper builds on the findings of Obleness et al (2012, G3) that showed expanded Olduvai triplets are responsible for the extreme increase in Olduvai copy number in humans. It would help to make sure that reference is credited for that observation.
3. The linking of Olduvai copy number and human brain evolution was first suggested in Popesco et al 2006. It would be helpful if that point was made more explicit.
4. Major point: It is puzzling and surprising that the authors report no variation in Olduvai triplet copy number among various human genomes. This is in marked contrast to what was reported in other studies (for example, see Heft, et al 2020). This inconsistency should be addressed.
5. A recent study indicates that human NBPF proteins that contain many tandemly repeated Olduvai triplets are posttranslationally cleaved by the furin protease once at each triplet (Pacheco et al 2022). This indicates that Olduvai triplet proteins are the ultimate functional unit for Olduvai in humans. The authors may want to add this point to the paper since it adds more biological significance to a main part of their paper.
6. The quality of the writing suffers from poor English grammar in numerous parts of the paper. These should be addressed and corrected.
7. The font size in figure 1 varies at places where it should not.

Reviewer #2 (Comments to the Authors (Required)):

The paper is better reorganized and rewritten respect to the previous draft. I would suggest to accept it for publication in the present form.

September 30, 2022

RE: Life Science Alliance Manuscript #LSA-2021-01306-TRR

Prof. Matko Gluncic
University of Zagreb
Faculty of Science
Bijenicka cesta 32
Zagreb 10000
Croatia

Dear Dr. Gluncic,

Thank you for submitting your Research Article entitled "Tandemly repeated NBPF HOR copies (Olduvai triplets) - possible impact on human brain evolution". It is a pleasure to let you know that your manuscript is now accepted for publication in Life Science Alliance. Congratulations on this interesting work.

DISTRIBUTION OF MATERIALS:

Again, congratulations on a very nice paper. I hope you found the review process to be constructive and are pleased with how the manuscript was handled editorially. We look forward to future exciting submissions from your lab.

Sincerely,
